# Characterization of the Complete Mitochondrial Genome of Wintersweet (*Chimonanthus praecox*) and Comparative Analysis within Magnoliids

**DOI:** 10.3390/life14020182

**Published:** 2024-01-25

**Authors:** Xianxian Yu, Yanlei Feng, Jie Zhang

**Affiliations:** 1College of Urban and Environmental Sciences, Xuchang University, Xuchang 461000, China; yuxianxian2016@163.com; 2Hangzhou Global Scientific and Technological Innovation Center, Zhejiang University, Hangzhou 311200, China; fengyanlei@zju.edu.cn; 3Lushan Botanical Garden, Chinese Academy of Sciences, Jiujiang 332900, China

**Keywords:** mitogenome, *Chimonanthus praecox*, magnoliids, comparative analysis

## Abstract

Mitochondrial genome sequencing is a valuable tool for investigating mitogenome evolution, species phylogeny, and population genetics. *Chimonanthus praecox* (L.) Link, also known as “La Mei” in Chinese, is a famous ornamental and medical shrub belonging to the order Laurales of the Calycanthaceae family. Although the nuclear genomes and chloroplast genomes of certain Laurales representatives, such as *Lindera glauca*, *Laurus nobilis*, and *Piper nigrum*, have been sequenced, the mitochondrial genome of Laurales members remains unknown. Here, we reported the first complete mitogenome of *C. praecox*. The mitogenome was 972,347 bp in length and comprised 60 unique coding genes, including 40 protein-coding genes (PCGs), 17 tRNA genes, and three rRNA genes. The skewness of the PCGs showed that the AT skew (−0.0096233) was negative, while the GC skew (0.031656) was positive, indicating higher contents of T’s and G’s in the mitochondrial genome of *C. praecox*. The Ka/Ks ratio analysis showed that the Ka/Ks values of most genes were less than one, suggesting that these genes were under purifying selection. Furthermore, there is a substantial abundance of dispersed repeats in *C. praecox*, constituting 16.98% of the total mitochondrial genome. A total of 731 SSR repeats were identified in the mitogenome, the highest number among the eleven available magnoliids mitogenomes. The mitochondrial phylogenetic analysis based on 29 conserved PCGs placed the *C. praecox* in Lauraceae, and supported the sister relationship of Laurales with Magnoliales, which was congruent with the nuclear genome evidence. The present study enriches the mitogenome data of *C. praecox* and promotes further studies on phylogeny and plastid evolution.

## 1. Introduction

*Chimonanthus praecox* (L.) Link. 1822 (commonly known as wintersweet or “La Mei” in Chinese) is a perennial deciduous shrub of the Calycanthaceae family [1]. It is an essential ornamental plant native to China and widely cultivated in bonsai cultivation, landscaping, and cut flower production because of its distinctive fragrant aroma and blossoming time (from late November to the following March). Thus, wintersweet is planted in almost all Chinese gardens and green spaces. It has also been widely utilized for centuries for its medicinal uses for treating rheumatism, measles, cough, and heatstroke, especially in traditional Chinese medicine [2,3]. Modern pharmacological research has found that wintersweet exhibits multiple therapeutic activities, such as antitumor, antiviral, and immunomodulatory [4,5].

*C. praecox* belongs to the Laurales, a part of Magnoliidae (magnoliids), also including the orders Piperales, Canellales, and Magnoliales [6,7]. Magnoliidae are the third-largest group of Mesangiospermae, containing approximately 9000 species. Some magnoliids share anatomical similarities with gymnosperms, such as carpels, which are situated on the flowering axis and are considered to be the early diverging lineages from angiosperms [8]. The analysis of the evolutionary problems of magnoliids helps us understand the evolution of angiosperms, which makes them occupy a pivotal position in the research of plant evolution.

Except for non-mesangiosperms (the ANA grade, i.e., Amborellales, Nymphaeales, and Austrobaileyales), magnoliids are the key nodes to reveal the evolution of angiosperm. An increasing number of magnoliids genomes (such as *Cinnamomum kanehirae*, *Liriodendron chinense*, *Magnolia biondii*, *Chimonanthus praecox*, *Chimonanthus salicifolius*, *Aristolochia fimbriata*, *Piper nigrum*, and *Phoebe bournei*) facilitate the studies of flowering plant evolution and the mechanism of many important traits. For the past few decades, scientists have been grappled with the angiosperms’ origin and evolution problems [9,10,11]. However, the evolutionary problems of magnoliids remain controversial. For example, the analysis of the recently released genome of magnoliids (*Piper nigrum*, *Liriodendron chinense*, *Cinnamomum kanehirae*, and *Persea americana*) results in two conflicting conclusions regarding their relationship with monocots and eudicots [12,13,14,15]. Frequent lineage-specific whole genome duplication (WGD) events have occurred throughout the evolutionary history of magnoliids. For instance, two rounds of ancient WGD in the genome of *Chimonanthus salicifolious*, three successive rounds of lineage-specific WGDs in black pepper, single round of WGD in *Liriodendron chinense*, and two rounds of WGD in *Cinnamomum kanehirae* were observed [10,16,17]. This resulted in rapid gene gain and loss, pseudogenization, and purifying selection, complicating the evolutionary history of homologous genes [18]. The rapid radiation and ancient lineage sorting hybridization observed in other angiosperms may partly explain the elusive problem of evolutionary issues in magnoliids [19,20,21].

Although more than 23 nuclear genomes from magnoliid species have been reported and great efforts have been taken to study the evolution of magnoliids using molecular data such as nuclear genome, plastid genome, or mitochondrial genome, the current understanding on magnoliids phylogeny is yet inconsistent [22,23]. Thus, mitochondrial genomes, housing the oxidative phosphorylation machinery and many other essential metabolic pathways, may provide additional sophisticated evidence for the phylogenetics of magnoliids [24]. Unfortunately, the mitochondrial genomes of many important taxa, such as Calycanthaceae, Monimiaceae, and Canellaceae, are not yet available.

As a vital ornamental and medicinal plant, wintersweet also plays a vital role in clarifying the evolution of angiosperms. The chromosome-level genome of *C. praecox* and the cultivar *C. praecox* cv. *Concolor* have been released, providing new insights into floral scent biosynthesis and flowering in winter [25,26,27]. However, the mitochondrial genomes of *C. praecox* and many species in Laurales are yet to be elucidated. The mitochondrial genome conventionally shares the similarities of conserved maternal inheritance with the chloroplast genome, but it has more complex with various structural features. For example, the mitochondrial genome usually consists of a single double-stranded circular DNA in many plants, similar to the chloroplast genome in *Arabidopsis thaliana* and *Cucurbita pepo*, while linear DNA and multiple circular DNA molecules have also been found in rice, wheat, and cucumber [28,29,30,31,32]. Plant mitogenomes vary greatly in size, normally ranging from 66 Kb to more than 10 Mb [33]. Meanwhile, they evolve rapidly in structure such that even close relatives have clear differences, such as in cotton [34]. These peculiarities make their assembly tough. Currently, there are approximately 700 sequenced complete mitogenomes, which is substantially less than the number of plastomes [35].

In the present study, we assembled and annotated the mitogenome of *C. praecox* to analyze its gene content, repetitive sequence, codon usage, and synonymous (Ks) and nonsynonymous (Ka) substitution. Comparative analysis with the available mitogenomes of magnoliids was also performed. It will be a valuable resource for further studies on the mitochondrial genome evolution and phylogeny of angiosperm.

## 2. Materials and Methods

### 2.1. Plant Materials, Library Preparation, and Genome Sequence

The fresh young leaves of a single *Chimonanthus praecox* plant were collected from plants in Bailiang Town (34°09′ N, 114°16′ E), Yanling County, Henan Province, China. A voucher specimen was deposited at Xuchang University, Xuchang City, Henan Province, China. Total genomic DNA was extracted using a modified CTAB (cetyl trimethyl ammonium bromide) method [36] and quality controlled using agarose gel electrophoresis and Nanodrop 2000 Spectrophotometer (Thermo Fisher Scientific, Wilmington, DE, USA). After quality testing, short-insert libraries within 350 bp were constructed using the standard manufacturer’s PCR-free protocol and sequenced on the Illumina HiSeq2500 platform (Illumina, San Diego, CA, USA).

### 2.2. Mitochondrial Genome Assembly and Annotation

The raw reads obtained with Illumina HiSeq2500 were trimmed, and the low-quality bases (<Q20) were filtered out using the NGS QC Toolkit (v2.3) [37]. Around four gigabase reads were used to assemble mitogenome with SPAdes v3.14.0 [38]. The contigs were visualized in Bandage v0.8.1 to remove the fragments from the chloroplast and nuclear genome [39]. The contigs were connected manually in Geneious R8 (Biomatters, Inc., Auckland, New Zealand). The new complete mitochondrial genomic sequence was annotated using the DOGMA (http://dogma.ccbb.utexas.edu/, accessed on 10 July 2022) online program [38]. Finally, the new sequence was available in GenBank with accession number OR811177. The circular mitogenome map of *C. praecox* was generated using OG-DRAW (https://chlorobox.mpimp-golm.mpg.de/OGDraw.html, accessed on 20 July 2022) [40]. The following formulas were used to assess mitogenome or PCGs’ strand asymmetry: AT skew = [A − T]/[A + T]; GC skew = [G − C]/[G + C].

### 2.3. Repetitive Sequence Analysis

The repetitive sequence of the mitogenome of *C. praecox* was detected using REPuter (https://bibiserv.cebitec.uni-bielefeld.de/reputer, accessed on 25 August 2022) with the following parameters: the maximum computed repeats of 5000, the minimum repeat size of 30, and a hamming distance of 3 [41]. Simple sequence repeat (SSR) was detected using the MISA online program (https://webblast.ipk-gatersleben.de/misa/, accessed on 11 September 2022) with SSR motif sizes of 1, 2, 3, 4, 5, and 6 bases with thresholds of 8, 4, 4, 3, 3, and 3 repeat numbers, respectively [42]. The mitogenome sequences of the other species were downloaded from the NCBI website (www.ncbi.nlm.nih.gov/genome/organelle/, accessed on 11 September 2022). The figures were constructed using the ggplot2 package in the R (v4.1.2).

### 2.4. Codon Usage Analysis

Relative synonymous codon usage (RSCU) was the ratio of times a particular codon was observed to the expected frequency of all synonymous codons of the same amino acid. If RSCU = 1, it indicates that codon usage is unbiased. If RSCU < 1 or > 1, it indicates that the actual frequency of use of the codon is lower or higher than that of other synonymous codons, respectively [43]. The mitogenome’s total codon distribution (count) and RSCU were analyzed in MEGA11 [44] and CodonW (version 1.4.2, http://codonw.sourceforge.net/, accessed on 10 November 2023). All synonymous codons were identified to find RSCU values. A codon used less frequently than expected will have an RSCU value of less than one, and vice versa for a codon used more frequently than expected.

### 2.5. Synonymous and Nonsynonymous Substitution Ratio

The sequences of 27 shared PCGs of *C. praecox* and the other eight species were aligned separately using the MAFFT v7.520 [45]. The nonsynonymous (Ka) and synonymous (Ks) substitution ratio (Ka/Ks) of the 27 mitochondrial PCGs were analyzed using DnaSP v6.12.03 to identify the genes that are under selection pressure [46]. Statistical analysis of the Ka/Ks ratio of 27 protein-coding genes was performed with one-way ANOVA using R v4.1.2.

### 2.6. Phylogenetic Analysis

Since a large number of phylogenetic findings support magnoliids (consisting of Magnoliales, Laurales, Canellales, and Piperales) as a monophyletic branch of Mesangiospermae, we searched and downloaded all available mitochondrial genomes in magnoliids from NCBI database. (https://www.ncbi.nlm.nih.gov/genbank/, accessed on 10 November 2023) for phylogenetic analysis [22,47,48].

To determine the phylogenetic position of *C. praecox*, the coding sequence (CDS) of the 29 conserved mitochondrial PCGs (atp1, atp4, atp6, atp8, atp9, ccmB, ccmC, ccmFc, ccmFn, cox1, cox2, cox3, cytb, matR, mttB, nad1, nad2, nad3, nad4, nad4L, nad5, nad6, nad7, nad9, rps1, rps3, rps4, rps12, and rps13) presented in these mitogenome were extracted in PhyloSuite v1.2.3 [49]. The sequences of the 29 PCGs were aligned separately using Muscle and trimmed with an automated1 option of trimAl, which was implemented in PhyloSuite [49,50]. The trimmed sequences were concatenated and used to construct a maximum likelihood (ML) tree in IQ-TREE (v1.612) with the following parameters: -m MFP -bb 1000, and setting the *Nelumbo nucifera* as an outgroup [51]. The bootstrap values in each clade of the phylogenetic tree were calculated based on 1,000 replicates. The best evolutionary model was automatically selected as ‘GTR + F + R2’ according to the Bayesian Information Criterion (BIC) value generated by ModelFinder in IQ-TREE. The phylogenetic tree was visualized with the iTOL online program (https://itol.embl.de/, accessed on 10 November 2023).

## 3. Results and Discussion

### 3.1. Mitogenome Structure, Organization, and Composition

A total of 4.03 gigabase raw reads were generated using Illumina HiSeq2500 (Illumina, San Diego, CA, USA). The primary de novo assembly used SPAdes and generated 2903 contigs within a total length of 11,168,911 bp (Appendix A). Among them, 2649 contigs have a length greater than 1000 bp, with a median length of 2412 bp and a maximum length of 79,783 bp (Appendix A). After manually removing the redundant fragments, the mitogenome of *C. praecox* was assembled into a single circular DNA molecule with a total length of 972,347 bp after the gap-filling step (Figure 1). The size is close to that of *Magnolia biondii* (NC_049134.1, 967,100 bp) but nearly triple the size of *Aristolochia fimbriata* (OP649454.1-QP649456.1, 349,849 bp) (Appendix A).

The nucleotide composition of the mitogenome is A: 26.23%; T: 26.38%; C: 23.89%; and G: 23.51%. The AT and GC contents are 52.61% and 47.39%, respectively, which are similar to those of magnoliids species (Appendix A) [52]. Both the AT skew (−0.002851169) and GC skew (−0.008016878) were negative, indicating higher T and C in the mitogenome of *C. praecox*. The mitogenome encodes 60 unique genes, including 40 PCGs, 3 rRNA, and 17 tRNA (Table 1). The whole mitogenome sequence of *C. praecox* has been deposited in the GenBank database with an accession number of OR811177. Despite the significant variation in mitochondrial genome size, gene set and GC content were consistent in magnoliids [24,52].

### 3.2. Genomic Features of the C. praecox Mitogenome

Among the PCGs of the *C. praecox* mitogenome, five genes (*nad1*, *nad2*, *nad5*, *atp1*, and *rps19*) were duplicated two or more times. Notably, the presence and number of introns within these PCGs varied significantly. The majority of PCGs lacked introns entirely, while eight genes contained introns. Specifically, five genes (*nad1*, *nad2*, *ccmFc*, *rps3*, and *rps10*) had only one intron each, while the *cox2*, *nad4*, and *nad7* genes contained two, three, and four introns, respectively.

The combined length of all PCGs amounted to 35,827 bp, corresponding to approximately 3.7% of the total mitogenome length. The average length of PCGs was 1526 bp, spanning from 21 bp to 10,390 bp (Appendix A). The nucleotide composition of PCGs was A: 25.34%; T: 25.83%; G: 25.19%; and C: 23.64%, which resulted in an AT content of 51.17% and a CG content of 48.83%. The AT skew (−0.0096233) and GC skew (0.031656) indicated a higher T and G content than that of A and C among the PCGs.

Three ribosomal RNA genes and seventeen transfer RNA genes were identified in the mitogenome. The total length of rRNA genes was 9504 bp, which comprised the duplicated copies of *rrnL* and the single copies of *rrn5* and *rrnS* (Table 1, Appendix A). Seventeen unique tRNA genes were identified in the mitogenome of *C. praecox*, and ten genes have multiple copies. Specifically, two copies were found in eight tRNA genes (*trnD*, *trnE*, *trnF*, *trnK*, *trnP*, *trnQ*, *trnS*, *trnY*), while *trnL* had three copies, and *trnM* had five copies in the mitogenome of *C. praecox*. These tRNA genes ranged from 67 bp to 104 bp, totaling 2323 bp (Appendix A).

### 3.3. Codon Usage Analysis

Most amino acids are encoded by 2–6 different codons except for tryptophan and methionine, and codon usage bias reflects the mutation patterns and evolution of genes [53]. To better understand the mitogenome features of *C. praecox*, we performed a comparative analysis of the mitogenomes of *C. praecox* with those of other species, including *Ginkgo biloba*, *Magnolia biondii*, *Liriodendron tulipifera*, *Nelumbo nucifera*, *Oryza sativa* (*japonica*), *Aconitum kusnezoffii*, *Aristolochia fimbriata*, and *Saururus chinensis* [54,55,56,57]. The genome size and number of PCGs varied among species. For instance, *Magnolia biondii* and *Liriodendron tulipifera* are magnoliids with similar amounts of PCGs, rRNA, and GC percentage (Appendix A).

In the mitogenome of *C. praecox*, a total of 11,894 codons were identified in 40 PCGs except for the termination codons, and the majority of PCGs initiated with ATG start codons with a few exceptions (*nad1*, *nad2*, *nad4L*, *nad5*, *rps4*, and *rps10*). Notably, five types of stop codons (CGG, AAA, TAT, GTA, and GGT) were found in the mitogenome of *C. praecox* in addition to the traditional stop codons (TAA, TAG, and TGA). The codon usage analysis revealed that arginine (Arg), serine (Ser), and leucine (Leu) are the most common amino acid residues. At the same time, tryptophan (Trp) is the least-used amino acid residue in the mitogenome (Figure 2, Appendix A).

Comparing the mitogenome of *C. praecox* with the other eight species showed that the majority of amino acids are similar among all species except for alanine (Ala), cysteine (Cys), proline (Pro), methionine (Met), and Leu. The percentages of Ala, Met, and Pro in *C. praecox* were lower than those in the other species, while the proportion of Cys was more significant than those in the other species (Figure 3).

Relative synonymous codon usage (RSCU) analysis was conducted to assess codon usage bias. The RSCU analysis revealed that NNU and NNA were greater than one, with a few exceptions (AUA, CUA, AGU, and GUA). This pattern suggested a strong bias of A and T in the third position of codons in PCGs, which is also observed in the other plant mitogenomes [58,59]. An adequate number of codon (ENC) analysis was also performed to understand the effect of codon usage. The ENC value of the PCGs ranged from 39.23 to 60.99, indicating a very weak codon usage bias in these genes (Appendix A).

### 3.4. Synonymous and Nonsynonymous Substitution Ratio

The ratio of nonsynonymous and synonymous substitutions (Ka/Ks) of the PCGs was calculated to detect the selection pressure among the shared PCGs in the mitogenome of *C. praecox* and the other species. The Ka/Ks value was determined by analyzing the aligned sequences of the shared PCGs among nine species. The results revealed that the Ka/Ks value for the most shared PCGs was less than 1, which indicated that the synonymous substitution ratio exceeded the nonsynonymous substitution ratio. This result suggested that these genes were likely very conserved in evolution. For instance, only two genes (*nad2* and *rps7*) showed an average Ka/Ks ratio greater than one, suggesting that these genes had positive selection and tended to be retained or fixed during evolution (Figure 4). The remaining genes showed a Ka/Ks ratio of less than one, suggesting that these genes were likely to be highly conserved. For example, all Ka/Ks values in 11 genes (*atp1*, *atp6*, *ccmC*, *cox2*, *cox3*, *nad3*, *nad4L*, *nad5*, *nad7*, *nad9*, and *rps12*) were less than one, indicating that these genes were highly conserved and under purifying selection. To evaluate the negative selection of genes, one-way ANOVA was performed using *rps7* as the control, in which most values were close to one. In total, eight genes (*atp1*, *cob*, *cox1*, *nad1*, *nad2*, *nad4*, *nad5*, and *nad6*) showed statistically significant differences (*p*-value < 0.5, Appendix A).

### 3.5. Repetitive Sequence Analysis

The variation of mitogenome size in plants could be partly explained by different forms of repeats, such as composed tandem repeats, SSRs, or dispersed repeats that mainly include forward, reverse, complement, and palindromic sequences. The activation and prevalence of repeated sequences in plant mitogenomes play a pivotal role in remodeling the size and structure of plant mitogenomes [60,61]. Using the REPuter online program, a total of 2562 dispersed repeat sequences were found in the mitochondrial genome of *C. praecox* (Figure 5, Appendix A). The entire length of the dispersed repeat sequences was 165,069 bp, which accounted for 16.98% of the total genome size, and the sequence length ranged from 30 bp to 10,906 bp. The repeat number was greater than that in *L. tulipifera* (497), *O. sativa* (238), and *Magnolia biondii* (1295) but lower than that in *N. nucifera* (4759) and *G. biloba* (3529). The repeat length of the majority ranged from 30 bp to 100 bp with few large repeats. For instance, seven species (excluding *G. biloba* and *O. sativa*) contained large repeats of more than 10,000 bp, while *M. biondii* and *N. nucifera* contained more repeats of less than 200 bp. Notably, *O. sativa* contained the lowest quantity of repeats (238) but exhibited the most extended repeat size (45,584 bp). All the dispersed repeats in nine species were mainly classified into forward and palindromic types, and the number was similar in each species. The total length of repeats was also variable among species. Specifically, the total repeat length of *N. nucifera* was 301,185 bp, the largest of the nine species, accounting for 57.39% of the total genome size. A similar pattern was found in *G. biloba* and *O. sativa*, accounting for 39.89% and 30.74% of the total genome size, respectively. In the remaining species, the proportion ranged from 8.7% (*L. tulipifera*) to 17.88% (*S. chinensis*).

In addition to dispersed repeat, a total of 731 SSR repeats were identified in the mitogenome of *C. praecox*, including 228 mono-, 310 di-, 48 tri-, 116 tetra-, 22 penta-, and 7 hexa-nucleotide repeats (Figure 5, Appendix A). The total number of SSR was greater than that in *A. fimbriata* (278), *S. chinensis* (596), *M. biondii* (543), and *L. tulipifera* (517), which was the highest number among the nine species. This may partly explain the approximately 1 Mb (the largest among the nine mitogenomes) mitogenome of *C. praecox* (Appendix A).

### 3.6. Phylogenetic Analysis of C. praecox within Magnoliids

In our phylogenetic analysis, the mitochondrial sequences of *C. praecox* and the following twelve species were used: *Aristolochia fimbriata* (OP649454.1–OP649456.1), *Cinnamomum chekiangense* (NC_082065.1), *Cinnamomum insularimontanum* (OR176986.1), *Liriodendron tulipifera* (NC_021152.1), *Machilus pauhoi* (OR168698.1–OR168699.1), *Magnolia biondii* (NC_049134.1), *Magnolia figo* (NC_082234.1), *Magnolia liliiflora* (OR730816.1), *Magnolia officinalis* (NC_064401.1), *Nelumbo nucifera* (NC_030753.1), *Saururus chinensis* (OQ539548.1–OQ539550.1), and *Schisandra sphenanthera* (MH748549.1) (Appendix A). The sequence matrix for the phylogenetic analysis was 28,226 bp in length, accounting for 78.8% of the total length of protein-coding sequences in the *C. praecox* mitochondrial genome. A total of 1649 variable sites and 771 parsimony-informative sites (46.75% of the variable sites) were used. The phylogenetic tree placed the position of *C. praecox* within the Laurales (Figure 6).

Based on our phylogenetic relationships, the *C. praecox* forms a monophyletic clade with three other Lauraceae species (Figure 6). The phylogenetic tree formed three clades (Figure 6), of which clade I consisted of the species belonging to the order Magnoliales (*M. biondii*, *M. figo*, *M. liliiflora*, *M. officinalis*, and *L. tulipifera*), clade II consisted of species belonging to the order Laurales (*C. chekiangense*, *C. insularimontanum*, *M. pauhoi*, and *C. praecox*), and clade III comprised the species of the order Piperales (*A. fimbriata* and *S. chinensis*). Species from Laurales and Magnoliales further formed a clade with strong support (a bootstrap value of 100). Overlooking Canellales, which is the sister clade to Piperales, the topology of our phylogenetic tree is congruent with the chloroplast and nuclear genome evidence [22,48].

Unfortunately, only a few mitochondrial genome sequences in magnoliids are currently available. Only eleven magnoliids mitochondrial genomes were downloadable from the NCBI website, which is even less than the number of nuclear genomes that have been sequenced [23]. The eleven mitochondrial genomes belong to three orders (Piperales, Magnoliales, and Laurales, lacking Canellales) and five families (Saururaceae, Aristolochiaceae, Magnoliaceae, Calycanthaceae, and Lauraceae), accounting for only 27.78% of the total family number within magnoliids. Therefore, the mitochondrial sequencing of more representatives in magnoliids is necessary to explore their phylogenetic relationships.

## 4. Conclusions

This study assembled the complete mitogenome of *C. praecox* and performed its detailed analyses. The mitogenome of *C.praecox* was 972,347 bp in length with a GC content of 47.39% and 731 SSR repeats and comprised 40 PCGs, 17 tRNA genes, and 3 rRNA genes. The AT skew was negative, while the GC skew was positive. The Ka/Ks value of most genes was lower than one, indicating the purifying selection of these genes. Comparative analyses showed that the mitochondrial gene set, GC content, and codon usage of *C. praecox* mitogenome were consistent with those in magnoliids. The phylogenetic analysis using 29 shared protein-coding genes from eleven magnoliids mitochondrial genomes showed that *C. praecox* formed a monophyletic clade with three other Lauraceae species, and species from Laurales and Magnoliales formed a clade with strong support. Further, to provide more details on the evolution of the plant mitogenomes in non-mesangiosperms, more mitochondrial genomes are necessary.

## Figures and Tables

**Figure 1 life-14-00182-f001:**
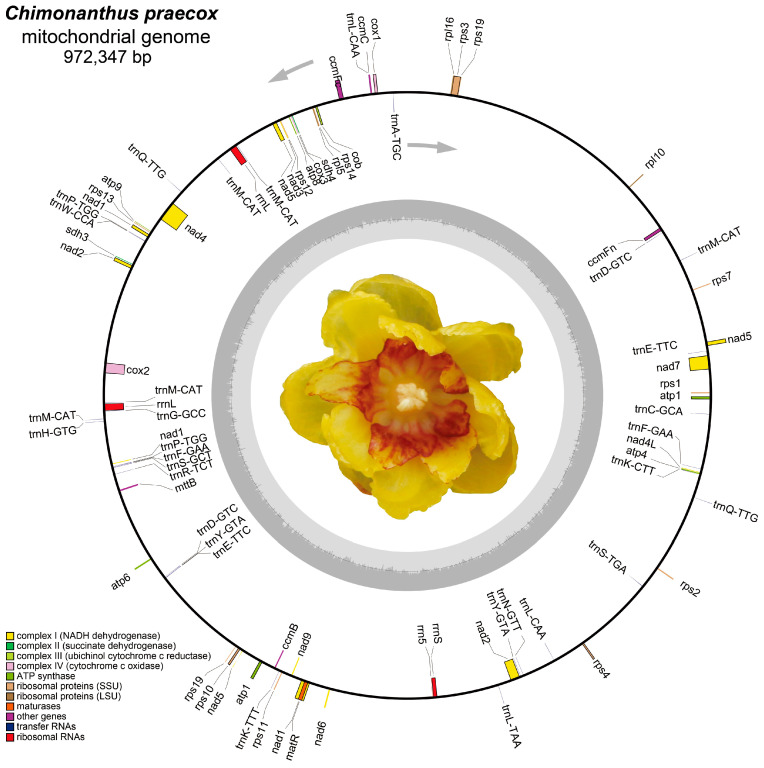
The complete mitogenome of *C. praecox*. Based on gene functions, the genes are represented as color bars in the circle. Genes represented outside the circle are transcribed counterclockwise, whereas those genes inside are transcribed clockwise, as shown by the gray arrow. The gray bars in the light gray circle indicate the percentage of GC content.

**Figure 2 life-14-00182-f002:**
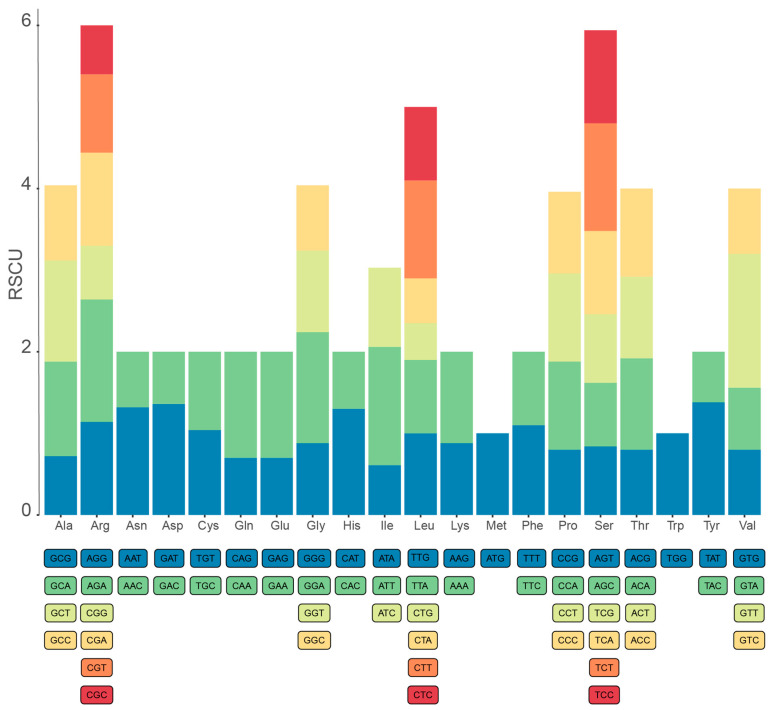
Codon usage analysis of protein-coding genes (PCGs) in the mitogenome of *C. praecox*. Relative synonymous codon usage (RSCU) values are plotted on the *y*-axis.

**Figure 3 life-14-00182-f003:**
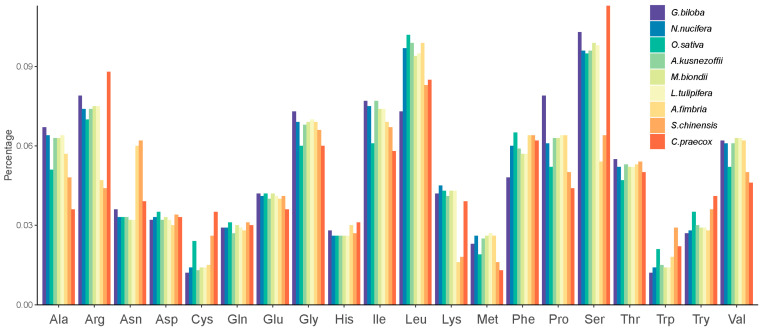
Relative amino acid percentage of *C. praecox* compared with the other six species. The *y*-axis indicates the percentage of amino acids in the CDS in each species.

**Figure 4 life-14-00182-f004:**
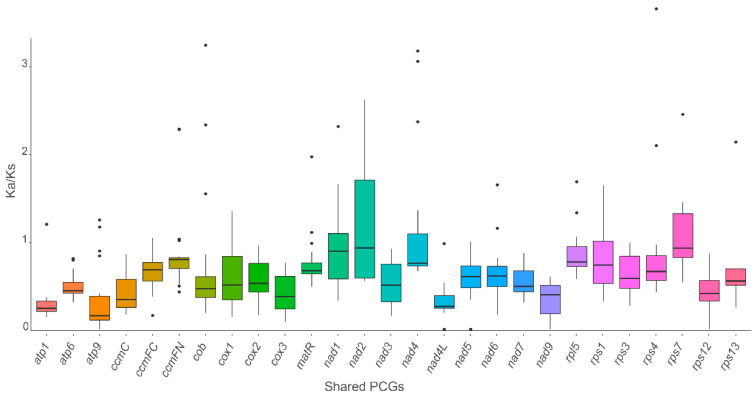
Box plot for pairwise divergence nonsynonymous and synonymous ratio (Ka/Ks) for shared PCGs of the nine mitogenomes. *y*-axis indicates the Ka/Ks value, while the *x*-axis indicates the shared PCGs. Points indicate the outliers.

**Figure 5 life-14-00182-f005:**
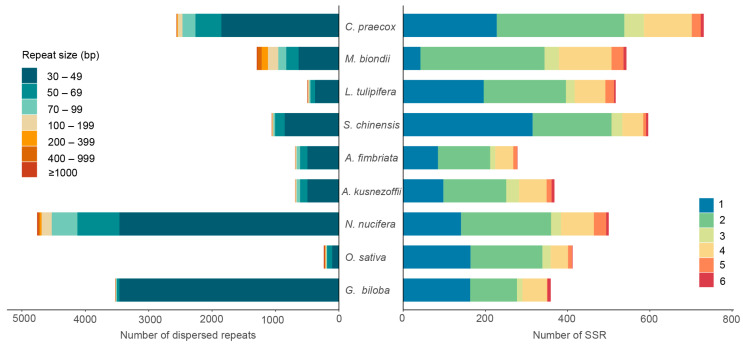
The repetitive sequence analysis. The figure on the left shows the distribution of dispersed repeats in nine species, while the figure on the right shows the distribution of SSRs. Different colors in the legend of the left figure represent the repeat size. In contrast, in the right figure, numbers in legend (1–6) correspond to mono-, di-, tri-, tetra-, penta-, and hexa-nucleotide repeats of SSR sequences, respectively. The *x*-axis indicates the number of dispersed repeats (left) or SSRs (right).

**Figure 6 life-14-00182-f006:**
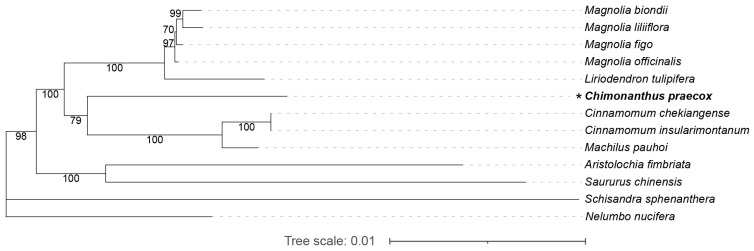
Phylogenetic relationships of *C. praecox* and other ten available magnoliids species. The unrooted evolutionary tree was reconstructed using the concatenated sequences of 29 protein-coding genes in 13 mitochondrial genomes using the maximum likelihood method. Bootstrap support values from 1000 replicates are shown above branches. The new mitochondrial genome obtained in this study is marked with a star.

**Table 1 life-14-00182-t001:** The gene present in the mitochondrial genome of *C. praecox*.

Group of Genes	Gene Names
Complex I (NADH dehydrogenase)	*nad1* ^a^, *nad2* ^a^, *nad3*, *nad4* ^ad^, *nad4L*, *nad5* ^a^, *nad6*, *nad7* ^ad^, *nad9*
Complex II (succinate dehydrogenase)	*sdh3*, *sdh4*
Complex III (ubiquinol cytochrome c reductase)	*cob*
Complex IV (cytochrome c oxidase)	*cox1*, *cox2* ^ac^, *cox3*
Complex V (ATP synthase)	*atp1* ^b(2)^, *atp4*, *atp6*, *atp8*, *atp9*
Cytochrome c biogenesis	*ccmB*, *ccmC*, *ccmFc* ^a^, *ccmFn*
Ribosomal proteins (SSU)	*rps1*, *rps2*, *rps3* ^a^, *rps4*, *rps7*, *rps10* ^a^, *rps11*, *rps12*, *rps13*, *rps14*, *rps19* ^b(2)^
Ribosomal proteins (LSU)	*rpl5*, *rpl10*, *rpl16*
Maturases	*matR*
Transport membrane protein	*mttB*
Ribosomal RNAs	*rrn5*, *rrnL* ^b(2)^, *rrnS*
Transfer RNAs	*trnA*, *trnC*, *trnD* ^b(2)^, *trnE* ^b(2)^, *trnF* ^b(2)^, *trnG*, *trnH*, *trnK* ^b(2)^, *trnL* ^b(3)^, *trnM* ^b(5)^, *trnN*, *trnP* ^b(2)^, *trnQ* ^b(2)^, *trnR*, *trnS* ^b(2)^, *trnW*, *trnY* ^b(2)^

a—intron-containing genes; b—duplicated copies of the genes; the number of duplicates is represented in the bracket; c—gene containing two introns; and d—gene containing three or more introns. Apex notation indicates the number of copies.

## Data Availability

The whole mitogenome sequence of *C. praecox* has been deposited in the GenBank database with an accession number of OR811177.

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
