# Peer review of "Characterization of the Complete Mitochondrial Genome of Wintersweet (Chimonanthus praecox) and Comparative Analysis within Magnoliids"

_life, 2024, doi:10.3390/life14020182_

Round 1

Reviewer 1 Report

Comments and Suggestions for Authors

In the manuscript titled Characterization of the complete mitochondrial genome of wintersweet (Chimonanthus praecox) and comparative analysis within Magnoliids, Yu et al. assembled and annotated the mitogenome of C. praecox, researched on molecular evolution and phylogeny.

This study contains some interesting findings and are valuable for the understanding of the evolution of C. praecox. The research purpose is clear and the research method is comprehensive.  Minor revision has to be done before this manuscript could be accepted for publication in the life.

1 In the abstract section, the authors need to clearly point out the AT skew and GC skew in PCGs or the whole mitogenome (line 19), results of different intervals may give different conclusions.

2 In the introduction section, the authors need to provide more detailed information on current progress of plant mitochondrial genome.

3 It is suggested to add collinearity analysis between C. praecox and other species, including structural variation detection.

3 I guess the comma is mislabeled in Line 72, please check the manuscript carefully.

4 Species names and gene names need italics in Line 151, Line 152 and Figure6, please check the manuscript carefully.

5 What do the light and dark gray parts in Figure1 represent? It is suggested to add pictures of C. praecox to facilitate readers to know the object of this study

6 In order to better understand the codon usage of C. praecox, it is suggested to add the codon usage of other species used in the article (Figure3), so that readers can better compare the codon usage of different species.

Author Response

Dear reviewer,

Thank you very much!

Reviewer 2 Report

Comments and Suggestions for Authors

Dear Authors,

Reviewer comments life-2787754

The manuscript entitled „Charcterization of the complete mitochondrial genome of wintersweet (Chimonanthus praecox) and comparative analysis within Magnoliids“ represents a useful study aimed at an investigation of the mitochondrial genome in wintersweet (Chimonanthus praecox) ornamental plant including mitochondrial genome sequencing, sequence analysis and its phylogenetic relationships to the published mitochondrial genomes in other Magnoliids. The manuscript provides data on the new complete mitochondrial genome sequence including the sequence analysis and comparison to other already published mitochondrial genome sequences in Magnoliids. I can therefore recommend the manuscript for publication in Life.

However, I have some comments on the present manuscript which should be addressed appropriately prior to the manuscript publication.

1/ I think that a separate Abbreviations listproviding an explanation of the less common terms such as „PGS“ for „protein-coding genes“ should be added to the manuscript.

2/ In Materials and methods, I think that the formulae used for the calculation of the ratio between synonymous and nonsynonymous substitutions in protein-coding genes (PCGs) have to be provided in the Materials and methods text or at least a relevant reference has to be given.

3/ Data availability statment providing the information on the public database where the complete mitochondrial genome sequence data of Chimonanthus praecox were deposited and the relevant accession number have to be provided.

4/ Formal comments on the text related to English language and style:

Line 226: Replace the word „less“ with „lower“ in the statement: „The percentages of Ala, Met, and Pro in C. praecox were lower than those in other species…“

Line 261: Replace the word „less“ with „lower“ in the statement: „The repeat number was greater than that in L. tulipifera (497), O. sativa (238), and magnolia biondii (1,295) but lower than that in N. nucifera (4,759) and G. Biloba (3,529).“

Line 266: Replace the word „the least“ with „the lowest“ in the statement: „Notably, O. sativa contained the lowest quantity of repeats (238)….“

Conclusions, line 318: Replace the word „less“ with „lower“ in the statement: „The Ka/Ks value of most genes was lower than 1 indicating teh purifying selection of these genes.“

Final recommendation: Accept after a minor revision.

Comments on the Quality of English Language

Dear Authors,

Reviewer comments life-2787754

The manuscript entitled „Charcterization of the complete mitochondrial genome of wintersweet (Chimonanthus praecox) and comparative analysis within Magnoliids“ represents a useful study aimed at an investigation of the mitochondrial genome in wintersweet (Chimonanthus praecox) ornamental plant including mitochondrial genome sequencing, sequence analysis and its phylogenetic relationships to the published mitochondrial genomes in other Magnoliids. The manuscript provides data on the new complete mitochondrial genome sequence including the sequence analysis and comparison to other already published mitochondrial genome sequences in Magnoliids. I can therefore recommend the manuscript for publication in Life.

However, I have some comments on the present manuscript which should be addressed appropriately prior to the manuscript publication.

1/ I think that a separate Abbreviations listproviding an explanation of the less common terms such as „PGS“ for „protein-coding genes“ should be added to the manuscript.

2/ In Materials and methods, I think that the formulae used for the calculation of the ratio between synonymous and nonsynonymous substitutions in protein-coding genes (PCGs) have to be provided in the Materials and methods text or at least a relevant reference has to be given.

3/ Data availability statment providing the information on the public database where the complete mitochondrial genome sequence data of Chimonanthus praecox were deposited and the relevant accession number have to be provided.

4/ Formal comments on the text related to English language and style:

Line 226: Replace the word „less“ with „lower“ in the statement: „The percentages of Ala, Met, and Pro in C. praecox were lower than those in other species…“

Line 261: Replace the word „less“ with „lower“ in the statement: „The repeat number was greater than that in L. tulipifera (497), O. sativa (238), and magnolia biondii (1,295) but lower than that in N. nucifera (4,759) and G. Biloba (3,529).“

Line 266: Replace the word „the least“ with „the lowest“ in the statement: „Notably, O. sativa contained the lowest quantity of repeats (238)….“

Conclusions, line 318: Replace the word „less“ with „lower“ in the statement: „The Ka/Ks value of most genes was lower than 1 indicating teh purifying selection of these genes.“

Final recommendation: Accept after a minor revision.

Author Response

Dear reviewer,

Thank you very much!

Reviewer 3 Report

Comments and Suggestions for Authors

The paper provides novel data to charcaterize at the molecular level a family of plants that was missing data on its mithocondrial genome. The research is well focised and falls into the scope of the present journal, but requires some minor improvemente before being acceptable.

The main problem is figure 5, where the legend is too concise and therefore is difficult to get the information. What does each point represents? Please define in the legend the meaning of PCG, which are the selected mitogenomes? The figures should be selfexplanotory, and this is very obscure. Please include a complete legend.

Figure 3: in the legend include "(RSCU)" after the definition, to explain the Y axis.

Figure 6: Use Bold type for "Figure 6", and "Magnolia liliflora" should be written in italics.

Lines 38: Change "chemistries" for "uses"

Line 72: delete "," and mind the space.

Author Response

Dear reviewer,

Thank you very much!
